# Pathogenicity and Pathotype Analysis of Henan Isolates of Marek’s Disease Virus Reveal Long-Term Circulation of Highly Virulent MDV Variant in China

**DOI:** 10.3390/v14081651

**Published:** 2022-07-27

**Authors:** Man Teng, Lu-Ping Zheng, Hui-Zhen Li, Sheng-Ming Ma, Zhi-Jian Zhu, Shu-Jun Chai, Yongxiu Yao, Venugopal Nair, Gai-Ping Zhang, Jun Luo

**Affiliations:** 1Key Laboratory of Animal Immunology, Ministry of Agriculture and Rural Affairs of China & Henan Provincial Key Laboratory of Animal Immunology, Henan Academy of Agricultural Sciences, Zhengzhou 450002, China; tm135@aliyun.com (M.T.); zhengluping_2006@126.com (L.-P.Z.); lihuizhen01@163.com (H.-Z.L.); mashengming66@163.com (S.-M.M.); luckzhuzhijian@163.com (Z.-J.Z.); chaishujun2008@163.com (S.-J.C.); 2UK-China Centre of Excellence for Research on Avian Diseases, Henan Academy of Agricultural Sciences, Zhengzhou 450002, China; 3International Joint Research Center of National Animal Immunology & College of Veterinary Medicine, Henan Agricultural University, Zhengzhou 450002, China; zhanggaip@126.com; 4College of Animal Science and Technology, Henan University of Science and Technology, Luoyang 471003, China; 5The Pirbright Institute & UK-China Centre of Excellence for Research on Avian Diseases, Pirbright, Ash Road, Guildford, Surrey GU240NF, UK; yongxiu.yao@pirbright.ac.uk (Y.Y.); venugopal.nair@pirbright.ac.uk (V.N.); 6Jiangsu Co-Innovation Center for Prevention and Control of Important Animal Infectious Disease and Zoonoses, Yangzhou University, Yangzhou 225009, China

**Keywords:** Marek’s disease, MDV, pathogenicity, oncogenicity, pathotype

## Abstract

In recent years, outbreaks of Marek’s disease (MD) have been frequently reported in vaccinated chicken flocks in China. Herein, we have demonstrated that four Marek’s disease virus (MDV) isolates, HN502, HN302, HN304, and HN101, are all pathogenic and oncogenic to hosts. Outstandingly, the HN302 strain induced 100% MD incidence, 54.84% mortality, and 87.10% tumor incidence, together with extensive atrophy of immune organs. Pathotyping of HN302 was performed in comparison to a standard very virulent (vv) MDV strain Md5. We found that both CVI988 and HVT vaccines significantly reduced morbidity and mortality induced by HN302 or Md5 strains, but the protection indices (PIs) provided by these two vaccines against HN302 were significantly lower (27.03%) or lower (33.33%) than that against Md5, which showed PIs of 59.89% and 54.29%, respectively. These data suggested that HN302 possesses a significant higher virulence than Md5 and at least could be designated as a vvMDV strain. Together with our previous phylogenetic analysis on MDV-1 meq genes, we have presently suggested HN302 to be a typical highly virulent MDV variant belonging to an independent Chinese branch. To our knowledge, this is the first report to provide convincible evidence to identify a pathogenic MDV variant strain with a higher virulence than Md5 in China, which may have emerged and circulating in poultry farms in China for a long time and involved in the recent MD outbreaks.

## 1. Introduction

Marek’s disease (MD) is one of the most serious avian immunosuppressive and neoplastic diseases of poultry [1]. MD is caused by Marek’s disease virus (MDV), an important oncogenic virus belonging to the family of alphaherpesviruses that induces rapid-onset T-cell lymphomas in its natural chicken hosts. Based on the antigenicity, MDV has been categorized into three serotypes: serotype 1 (MDV-1) or *Gallid alphaherpesvirus* 2 (GaHV-2), serotype 2 (MDV-2) or *Gallid alphaherpesvirus* 3 (GaHV-3), and serotype 3 (MDV-3), alternatively known as herpesvirus of turkey (HVT) that has been reclassified as *Meleagrid alphaherpesvirus* 1 (MeHV-1) [2]. Only the virulent MDV-1 strains are pathogenic and/or oncogenic and the virulence of different viruses varies greatly. Based on the mortality in chicken flocks, lesion frequency, and the levels immunoprotection in birds vaccinated with different currently used commercial MD vaccines, MDV-1 isolates have been further grouped into many pathotypes, designated as mild MDV (mMDV), virulent MDV (vMDV), very virulent MDV (vvMDV), and very virulent plus MDV (vv+MDV) [3,4]. 

In the past several decades, MD has caused huge economic losses to the poultry industry worldwide. MD is also the first example that demonstrated the use of vaccine immunization for preventing virally induced tumors [5]. Historically, three types of MD vaccines have been developed from strains belonging to the 3 MDV serotypes. These include the nonpathogenic vaccine strains of HVT FC-126 (serotype 3), MDV-2 SB-1 (serotype 2), and attenuated MDV-1 strains CVI988/Rispens and 814 (serotype 1), all of which have contributed immensely for the efficient control of MD [6,7]. However, vaccination can only prevent the MD tumor occurrences and clinical symptoms, but is unable to completely block virus infection, shedding, and transmission. Persistent immune pressure from continuous large-scale application of MD vaccines in chicken flocks is thought to have supported virus evolution towards increased virulence of MDV epidemic strains capable of breaking through the immunoprotection of the presently available MD commercial vaccines [8,9]. It is obvious that the emergence of such viruses with variant features have led to recent MD outbreaks in vaccinated chicken flocks, launching a new challenge for the effective prevention and control of MD in the future. 

In recent years, frequent outbreaks of MD cases in vaccinated chicken flocks have been reported in China, potentially suggesting the emergence of MDV pathotypes with increased virulence. In a previous study [10], we have reported the isolation of a total of 17 epidemic MDV strains from the MD vaccinated chicken flocks distributed in Henan province, central China. Phylogenetic analysis of the sequence of the MDV-specific Meq oncogene of these strains has demonstrated the prevalence of an independent genetic branch of Chinese MDV isolates, significantly distinct from those isolated from other geographical areas in North America, East Asia, Australia, and South Asia. More virulent MDVs with higher virulence or potential variants isolated from the chicken flocks immunized with HVT and/or CVI988 vaccines have also been reported in the provinces of Guangxi, Shanxi, Gansu, Shandong, Jilin, and Liaoning, from the south to north-east of China [11,12,13,14,15,16]. However, whether new vv+MDV or variant strains have emerged and widely circulated in Chinese chicken flocks and resulted in the frequent MD outbreaks still remains unclear. In order to provide convincing evidence for evaluating whether the vv+MDV or variant strains have been associated with the MD outbreaks in China, we investigated the pathogenicity and virulence of four Henan MDV isolates designated HN502, HN302, HN304, and HN101, and performed a pathotype analysis of HN302 using a similar method described by Avian Disease and Oncology Laboratory (ADOL) [3,4], with modifications in the choice of MD vaccines and lines of chickens for animal experiments. Our data have provided strong evidence, confirming the long-term circulation of Chinese MDV variants with at least a vvMDV virulence phenotype, which has broken through the immunoprotection provided by MD vaccines causing the recent outbreaks of MD in the chicken flocks in China.

## 2. Materials and Methods 

### 2.1. Viruses and Cells 

Four MDV-1 Chinese strains isolated from MD-vaccinated chicken flocks in Henan province, as listed in Table 1, were used for animal experiments in this study. The vvMDV strain Md5 (gift from Prof. Zhi-Zhong Cui, Shandong Agricultural University, China) served as the reference strain [17]. The vaccine strains of CVI988 and HVT/FC-126 were both isolated from commercial vaccines (Boehringer-Ingelheim, German) and passaged for two times. The primary chicken embryo fibroblast (CEF) monolayers were prepared from 9-day-old specific pathogen free (SPF) embryos (Beijing Boehringer Ingelheim Vital Biotechnology Co., Ltd., China), and the viral propagation and titration of all the MDV strains were performed on CEFs by plaque forming units (PFUs) measured as described previously [18]. 

### 2.2. Chickens 

Experiments were conducted in commercial SPF white Leghorn chickens (Spirax Ferrer Poultry Science and Technology Co. Ltd., Jinan, China) that were maintained in positive pressure-filtered air isolators in the animal facility.

### 2.3. Animal Experiments 

A total of 275 one-day-old SPF chickens were randomly divided into five groups (n = 55) and were housed in separate isolators. The birds in groups 1–4 were separately challenged with CEF-grown virus stocks containing 2000 PFUs of HN502, HN302, HN304, or HN101 viruses (Table 1), via the intra-abdominal inoculation at 1 day of age. The birds of group 5 were inoculated with an equal volume of mock CEF suspension and served as negative controls. Post-virus challenge, birds were daily observed for development of clinical MD symptoms, including the paralysis of legs, head and neck rotation, blind eyes, tumor occurrence, and mortality. At the end of the 75-day experimental period, all surviving birds were humanely euthanized and necropsied to check the gross tumors in visceral organs, especially in those of liver, spleen, kidney, and proventriculus. Except for the birds sacrificed for sample collection and early death before 14 days post-challenge (dpc), possibly due to the adverse effects of intra-abdominal infection, all the dead and survived birds showing MD symptoms and/or gross tumors were classified as cases of MD. The pathogenicity and oncogenicity of viruses were evaluated using the rates of cumulative morbidity, mortality, and tumor incidence.

For the evaluation of the pathotype of HN302, a total of 219 one-day-old SPF chickens were randomly divided into six groups (n = 37, 36, 36, 35, 37, or 38) and kept in six separate isolators for the 91-day experiments. At 1 day of age, birds from each of two groups were vaccinated by subcutaneous injection of 200 μL of HVT or CVI988 virus stocks (2000 PFU per bird). For the unvaccinated negative controls, birds in two groups were injected with the same doses of mock CEFs. At 7 days of age, all the birds from each group of HVT/CVI988-vaccinated or unvaccinated birds were separately challenged by Md5 or HN302 viruses (1000 PFU per bird). The birds were daily inspected for MD clinical symptoms and death. At the end of 91 days, all the surviving birds were humanely euthanized and examined to check the occurrence of gross tumors. Except for the early death before 14 dpc, the rates of cumulative morbidity, mortality and tumor incidence were exactly calculated as described above.

### 2.4. Determination of Body and Organ Weights 

To evaluate the effects of MDV infection on the bird’s growth, each of the five birds from virus-challenged or mock control groups were randomly selected for measuring their body weights at 7, 14, and 21 dpc. Then, these birds were humanely sacrificed for collecting the thymus and bursa of Fabricius. The relative weight of these lymphoid organs was determined by calculating the ratio of the weight of lymphoid organs (g) divided by the body weight of bird (g) multiplied by 100. 

### 2.5. Statistical Analysis 

The vaccine protective index (PI) was calculated as PI = {(% MD in unvaccinated chickens − % MD in vaccinated chickens)/% MD in unvaccinated chickens} × 100. The morbidity, percentage of MD incidence (% MD), body weight, lymphoid organ weight index, and survival data were analyzed using GraphPad Prism Version 6.0 (GraphPad Software, Inc., San Diego, CA, USA). Survival levels between the two groups were compared by Log-rank (Mantel–Cox) test. The significant differences in morbidity, mortality, tumor incidence, and the protection index (PI) between each group were analyzed by SPSS Chi-Square. Differences were considered to be statistically significant at *p* < 0.05.

## 3. Results 

### 3.1. Growth Rates and Lymphoid Organ Weights of Birds Challenged by Different MDV Strains 

To primarily compare the pathogenicity of four Chinese MDV-1 strains isolated from MD-vaccinated chicken flocks, we first determined the growth rates of birds that were separately challenged by viruses HN502, HN302, HN304, or HN101 (Table 1). As demonstrated in Figure 1a, no significant difference on body mass was observed between any of the groups of virus-challenged birds and mock controls in the first three weeks post-challenge. It is interesting that among different groups, although both of the bursa/body mass and thymus/body mass ratios were variable and only a part of them were lower in groups of the virus-challenged birds than that of the mock groups, no significant difference was observed at 7 or 14 dpc (Figure 1b,c). However, at 21 dpc, all the bursa/body mass ratios of the virus-challenged birds were lower than that of the control birds (Figure 1c), especially for the HNLC302-challenged birds showing a significant statistical difference (*p* < 0.01). 

**Table 1 viruses-14-01651-t001:** Background information of MDV strains used in this study.

No.	Strain Abbreviation	Original Signatures [10]	Year and Month	Source ^a^	Host	Ages for Virus Isolation (days)	Passages on CEF
**1**	HN502	HNLC502	2011, November	LC	Huangshan Yellow	145	7
**2**	HN302	HNLH302	2011, October	LH	Hyline Brown	120	11
**3**	HN304	HNLH304	2011, October	LH	Hyline Brown	120	8
**4**	HN101	HNJZ101	2016, June	JZ	Spotted chicken	120	3

^a^ LC, Luanchuan; LH, Luohe; JZ, Jiaozuo.

### 3.2. Variable Pathogenicities and Oncogenicities of Four Chinese MDV-1 Strains 

Except for the early non-specific death occurred in the first two weeks due to the adverse effects from intra-abdominal infection (n = 3, 9, 3, 8, or 5 for groups of HN502, HN302, HN304, HN101, and CEF, respectively) and birds sacrificed for collection of lymphoid organs during the first three weeks post challenge, all the remaining birds in each group were inspected for the evaluation of virus pathogenicity and oncogenicity. As shown in Table 2, the rates of cumulative morbidity, mortality and tumor incidence in virus-challenged birds are very different among four MDV strains. At the end of the 75-day experiments, the strain HN304 showed a lower morbidity (13.51%) and mortality (5.41%), together with the lowest gross tumor occurrence (8.11%). Increased pathogenicity and oncogenicity were observed for HN101 and HN502 strains, which displayed higher morbidities (21.88% and 35.14%), mortalities (6.25% and 13.51%), and tumor incidences (15.63% and 27.03%). HN302 virus caused the highest rates of 100% morbidity, 54.84% mortality, and 87.10% tumor incidence, respectively. These data indicated increased virulence among the different isolates, with the virulence rank increasing in order for the HN304, HN101, HN502, and HN302 isolates, in accordance with the survival curves observed in the virus-challenged birds (Figure 2).

### 3.3. Comparison of the Virulence of HN302 and Md5 in MD-Unvaccinated Birds 

To evaluate the potential virulence of the most virulent Chinese isolate HN302, a 91-day long-term post-challenge experiment was first performed to compare the MD incidence in birds infected with HN302 and Md5, prototype vvMDV reference strain. As shown in Table 3, both of HN302 and Md5 caused 100% morbidities of MD cases in the unvaccinated birds, with a mortality of 64.86% or 80.56%, respectively. The two MDV-1 strains of HN302 and Md5 also induced similar cumulative gross tumor incidences, 54.17% and 62.07%, respectively, in dead birds and both 100% in surviving birds, with an overall occurrence of 70.27% or 69.44%, respectively (Table 4). Although no significant difference of the cumulative morbidity, mortality, or gross tumor incidences was observed between the mock groups challenged by HN302 or Md5 viruses, the survival rate curves demonstrated that, compared to the rapid-onset death in Md5-challenged birds, the HN302-challenged birds displayed a higher survival rate and a slower onset of death (Figure 3). However, despite these differences, infection with both strains resulted in a 100% gross tumor occurrence in in birds surviving at the end of the animal experiment.

### 3.4. Pathotyping of HN302 in MD Vaccinated Birds Suggests It as a vv+MDV Strain 

To further characterize the pathotype of HN302, pathogenicity and oncogenicity of HN302 and Md5 strains were compared in MD-vaccinated chickens. As shown in Table 3 and Table 4, vaccination of birds with HVT or CVI988 significantly reduced the pathogenicity and oncogenicity compared to that of mock control group. The cumulative morbidity, mortality, and gross tumor incidence in HN302-challenged birds were reduced from 100%, 64.8%, and 70.27%, respectively, in the control group to 66.7%, 11.11%, and 61.11% in HVT-vaccinated birds and to 72.97%, 32.43%, and 62.16% in CVI988-vaccinated birds. The corresponding data for Md5-challenged birds were reduced from 100%, 80.56%, and 69.44%, respectively, for the control group to 45.71%, 22.86%, and 31.43% in HVT-vaccinated birds and to 42.11%, 28.95%, and 13.16% in CVI988-vaccinated birds, respectively. Our results also showed that none of the vaccine strains, HVT or CVI988, could provide complete protection against the two virulent viruses. As demonstrated in Table 3, protection indexes (PIs) of HVT-vaccination for HN302 or Md5-challenges were 33.33% and 54.29%, whereas for CVI988-vaccination, the corresponding PIs were 27.03% and 59.89%, with a significance of *p* < 0.05. In the HVT- or CVI988-vaccinated birds, both vaccine strains provided a better protective effect to block the tumor occurrence in Md5-challenged birds rather than those challenged by HN302 (*p* < 0.05) (Table 4). We found that in CVI988-vaccinated birds, HN302 had a significantly higher morbidity and tumor incidence but a significant lower PI than the vvMDV strain Md5. These data indicated that HN302 possesses a significant higher virulence than Md5; hence, it is reasonable to consider it a vv+MDV strain.

## 4. Discussion

In the past several decades, MDV has continually evolved in virulence, showing more severe disease in many parts of the world, especially in Asian countries [8]. Recently, failures of MD vaccination and occurrence of clinical disease have been frequently reported and a number of virulent MDV strains have been isolated and identified from MD-vaccinated chicken flocks in China [10,11,12,13,14,15,16]. Although the pathogenicity and oncogenicity of some of these MDV isolates have been determined through animal experiments, the pathotypes of most Chinese MDV isolates have not been determined and designated, mainly due to the lack of comparisons to standard v, vv, and vv+ MDV reference strains, e.g., GA, Md5, 648A, etc. Thus, we have presently used a similar method of MDV pathotyping described by ADOL [3,4], with modifications in the choice of vaccines and lines of chickens used in the animal experiments for the determination of the pathogenicity, oncogenicity, and pathotype grouping of the Henan MDV strains isolated from central China.

The first round of animal experiments using unvaccinated SPF White Leghorn chickens for virus challenge have showed that the four Henan MDV strains, HN502, HN302, HN304, and HN101, are all pathogenic and oncogenic to the birds. HN302 represented the strain with the highest pathogenicity and oncogenicity, with the virus-challenged birds displaying a 100% MD incidence, with high mortality and gross tumor incidence of 54.84% and 87.10%, respectively. Compared to the other Henan MDV strains, HN302 also caused serious atrophy of lymphoid organs in birds, indicating the high immunosuppressive effects. In the second round of animal experiments, we have attempted to determine the virulence and pathotype of HN302 strain using HVT- and CVI988-vaccinated SPF White Leghorn chickens (as opposed to the line 15 × 7 chickens in the ADOL protocol), and have compared it to that of the standard vvMDV prototype reference strain Md5. The results demonstrated that both of the viruses, HN302 and Md5, have caused the same morbidity (100%) in unvaccinated birds, with similar high incidence of gross tumors of 70.27% and 69.44%, respectively. However, in HVT- and CVI988-vaccinated birds, although both of the MD vaccines significantly reduced the morbidity and mortality in the HN302 or Md5-challenged birds, the protection rates of HVT and CVI988 vaccinations against HN302 (33.33% and 27.03%) were lower or significantly lower than that of Md5 challenged birds (54.29% or 59.89%, respectively). Furthermore, the tumor occurrences induced by HN302 were significantly higher than that of Md5, which were also observed in HVT or in CVI988 vaccinated birds. Taken together, our data strongly demonstrate that the virulence of HN302 is significantly higher than Md5, arguing for its designation at least as a vvMDV strain. Whether the virulence of HN302 is similar to or even higher than that of vv+MDV strains needs further pathotyping analysis using the bivalent (HVT+SB-1) vaccine and comparing to a prototype vv+MDV isolate, such as 648A, according to the standard ADOL method [3,4].

From 2012 to 2022, a number of virulent Chinese MDV isolates, such as SD2012-1 [11], LCC, LLY, LTS [12], SX1301 [13], BS/15 [14], GX18NNM4 [15], and LZ1309 [16], have been reported to have broken the protection provided by commercial MD vaccines of HVT, HVT+SB-1, CVI988, and/or 814. Among these MDV strains, only the virulence of LZ1309 and BS/15 have been determined and demonstrated to be similar to the vvMDV reference strain Md5, while for the other Chinese MDV strains, none of their pathotypes have been characterized following the ADOL method, which is widely recognized throughout the world. In the present study, our data not only confirmed that the HN302 can break through the immunoprotection induced by HVT and CVI988 vaccines, but also demonstrated that HN302 is at least could be designated as a vvMDV strain, with a significantly higher virulence than vvMDV prototype Md5. Previously, comparative sequence analysis of MDV oncogene meq has demonstrated that some amino acid mutations of MEQ proteins are common and consistent in virulent MDV strains and have been proven to be associated with increased virulence, while the other relatively consistent mutations have made the Chinese isolates to be new MDV variants belonging to an independent branch [8,9,10], which may have some similar biological characteristics that deserve to be further studied. In a previous study [14], some scholars have found that in the unvaccinated SPF chickens, some Chinese isolates displayed a late onset and low mortality compared to the control MDV strains. Presently, we have also found a similar phenomenon in that, compared to Md5, the acute death and rapid-onset tumor occurrence happened more slowly in HN302 challenged birds. Obviously, the delayed onset time and low early mortality of the disease caused by newly emerged highly virulent MDV variants may be more harmful to Chinese poultry industry, resulting in more economic losses for a longer time costs of feeding, vaccines, drugs etc. Considering that HN302 was isolated from the MD vaccinated chicken flocks as early as in 2011 [10], it is very likely that the newly emerged highly virulent MDV variant strains are circulating in the Chinese chicken flocks at least more than ten years, possibly accounting for the recent MD outbreaks in China.

It is well known that in the worldwide CVI988 vaccine has been regarded as a better choice for the efficient control of MD, but some of the vaccine strains could not provide ideal immune protection for some of the Chinese epidemic strains, especially for the highly virulent MDV variant strains such as HN302, as well as for recently reported MDV strain GX18NNM4 [15]. In China, a total of at least 191 farms distributed in 17 provinces have reported MD outbreaks between 2011 and 2021, and nearly 60 MDV epidemic strains have been isolated, most of which were isolated from CVI988/Rispens vaccinated chickens [8]. At present, under the vaccination pressure, several MDV variants have appeared in China and have acquired resistance to the MD vaccine-mediated immunity. The available commercial MD vaccines can only provide incomplete or poor immune protections against these MDV strains, implying that these viruses are probably still spreading on farms and will cause greater economic losses in the future. As one of the largest poultry industry bases in the world, it is urgent to develop a new generation of the MD vaccine for efficient control of the current prevalent Chinese MDV epidemic strains. With the latest generation and wide application of new technologies e.g., CRISPR/Cas9-based gene editing in vaccine development and the recent achievements in the development of new poultry vaccines [19,20,21,22], we are looking forward to the early generation of new efficient MD vaccines as soon as possible in the coming future.

## Figures and Tables

**Figure 1 viruses-14-01651-f001:**
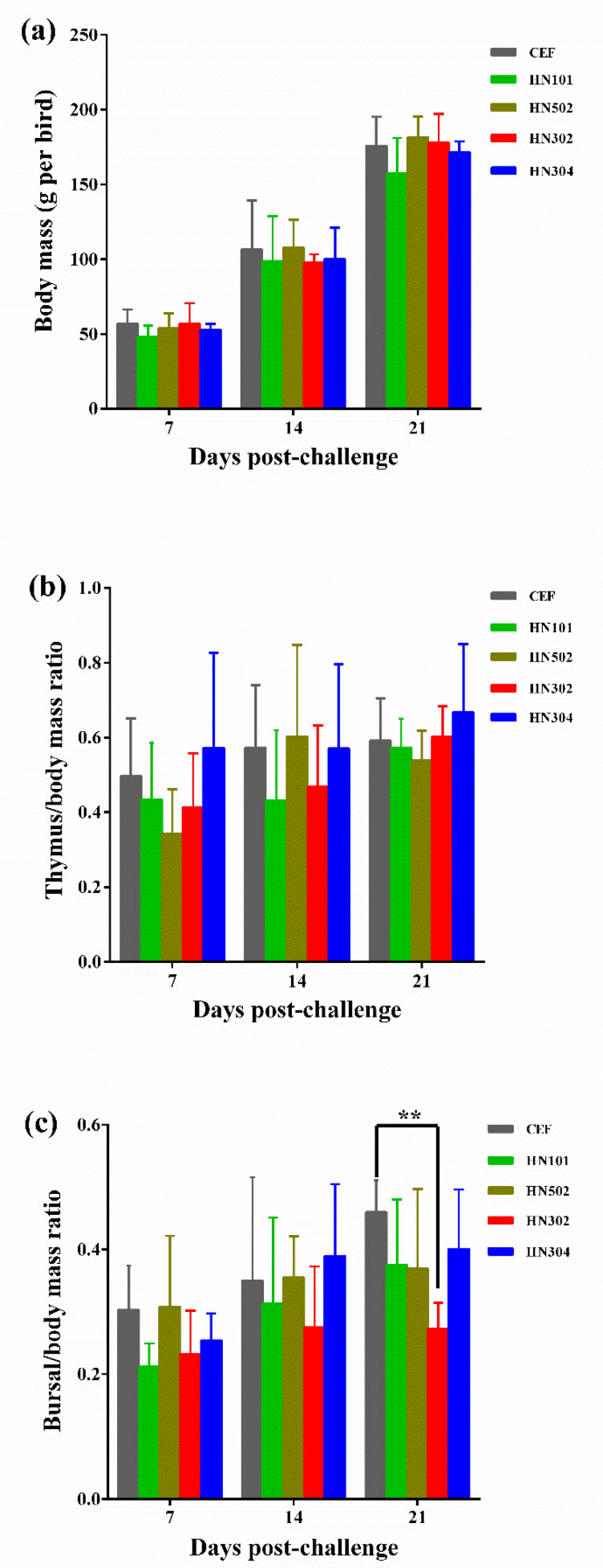
Growth rates and the ratios of bursa or thymus over body mass of chickens challenged with distinct MDVs. (**a**) Growth rates of virus-challenged birds. (**b**,**c**) Ratios of thymus or bursa over body mass of birds. For each group, the body masses, bursa masses, and masses of five randomly selected birds were measured in the first three weeks post-challenge. Black double stars indicate significant difference (*p* < 0.01) to CEF-mock controls.

**Figure 2 viruses-14-01651-f002:**
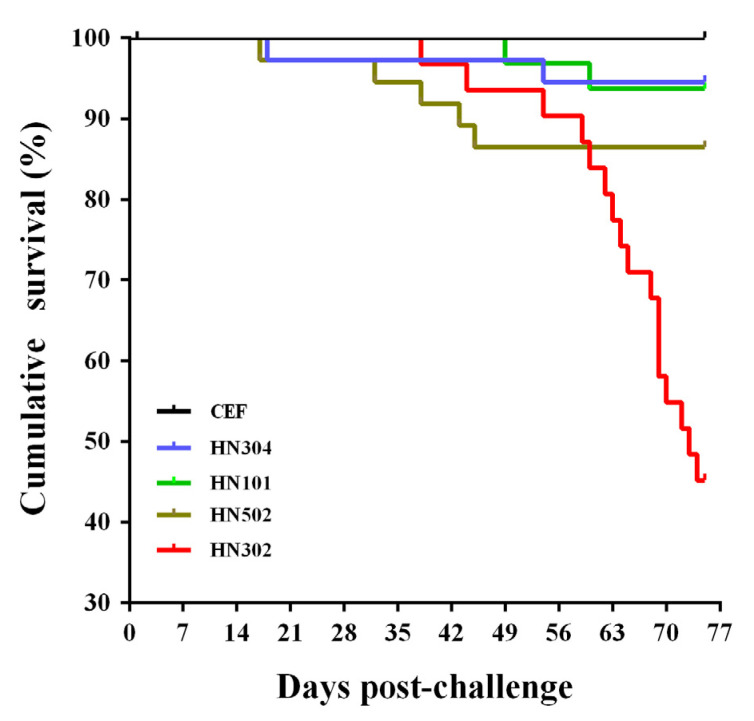
Survival curves of chickens challenged with distinct MDV strains over the 75-day experimental time period. For each experimental group, the birds were separately challenged with CEFs containing 2000 PFU viruses by intra-abdominal inoculation, while for the negative controls, a fifth group of birds were inoculated with an equal volume of mock-infected CEFs. Early deaths due to intra-abdominal infection adverse effects and birds sacrificed for body weighting and immune organ collections were excluded for data calculation.

**Figure 3 viruses-14-01651-f003:**
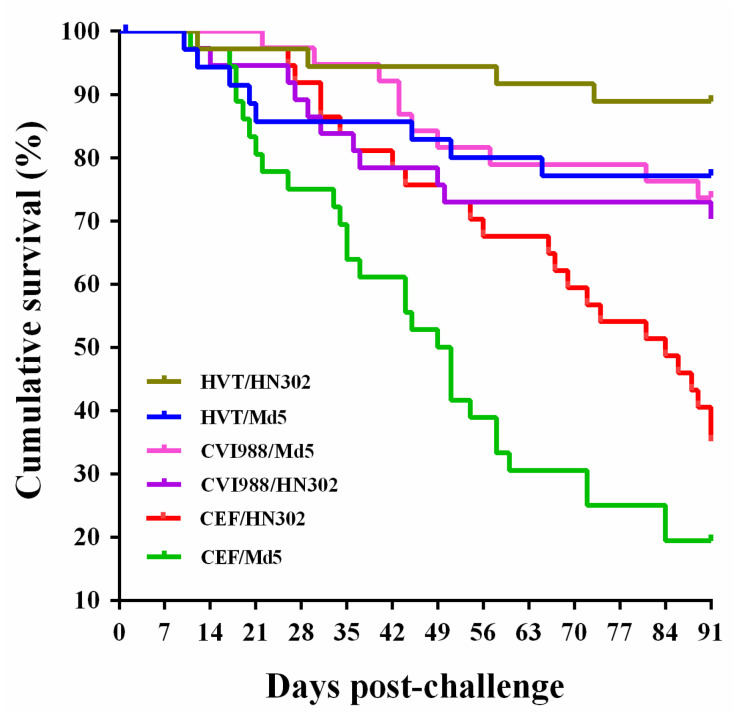
Survival curves of the MD-vaccinated and unvaccinated birds challenged with viruses HNLH302 or Md5 over the 91-day experimental time period. For each of two experimental groups, 1-day-old birds were separately vaccinated with CEFs containing 2000 PFU of HVT or CVI988 viruses by subcutaneous injection, while for the positive controls, two groups of birds were vaccinated with an equal volume of mock-infected CEFs. At 7 days of age, each of MD-vaccinated or mock control groups were separately challenged by HNLH302 or Md5 viruses (1000 PFU per bird) by intra-abdominal inoculation. Early deaths due to intra-abdominal infection adverse effects and birds sacrificed for body weighting and immune organ collections were excluded for data calculation.

**Table 2 viruses-14-01651-t002:** Cumulative morbidity, mortality, and gross tumor incidence in birds challenged by different MDV strains calculated at 75 days post-challenge (dpc).

No.	Strains	Total Numbers ^a^	Diseased Birds	Morbidity (%) ^b^	Deaths	Mortality (%) ^c^	Gross Tumors	Tumor Incidence (%) ^d^
**1**	HN502	37	13	35.14	5	13.51	10	27.03
**2**	HN302	31	31	100.00	17	54.84	27	87.10
**3**	HN304	37	5	13.51	2	5.41	3	8.11
**4**	HN101	32	7	21.88	2	6.25	5	15.63
**5**	CEF	35	0	0.00	0	0.00	0	0.00

^a^ Total number = number of virus-challenged birds in each group-total number of birds necropsied for weighting (15)-number of non-pathologically death. ^b^ All the birds showing clinical MD symptoms, death and survivals with gross tumors were classified as MD cases. Morbidity (%) = number of MD cases/total number × 100. ^c^ Mortality (%) = number of death/total number × 100. ^d^ Tumor incidence (%) = number of deaths or survivals with gross tumors/total number × 100.

**Table 3 viruses-14-01651-t003:** Cumulative morbidity and mortality in vaccinated birds challenged by HNLC302 or Md5 strains calculated at 91 days post-challenge (dpc).

Vaccines *	Strains	Total Numbers	Deaths	Mortality (%)	Diseased Birds ^#^	Morbidity (%)	PI (%)
CEF	HN302	37	24	64.86	37	100.00	/
	Md5	36	29	80.56	36	100.00	/
HVT	HN302	36	4	11.11 ^a^	24	66.67 ^a^	33.33
	Md5	35	8	22.86 ^a^	16	45.71 ^a^	54.29
CVI988	HN302	37	12	32.43 ^a^	27	72.97 ^ab^	27.03 ^b^
	Md5	38	11	28.95 ^a^	16	42.11 ^ab^	59.89 ^b^

* Equal volume of mock CEF suspension serves as vaccine negative control. ^#^ Surviving birds showing gross tumors inspected at 91 dpc were also calculated as MD cases. ^a^ Indicates significant difference between HVT- or CVI988-vaccinated groups and CEF-mock groups challenged by the same virus. ^b^ Indicates significant difference between HNLH302 and Md5 viruses in the same vaccination group. “PI” means the protection index. “/” means not applicable.

**Table 4 viruses-14-01651-t004:** Cumulative gross tumor incidence in vaccinated birds challenged by HNLC302 or Md5 strains calculated at 91 days post-infection (dpi).

Vaccines *	Strains	Total Birds	Deaths		Survivals		Total	
			Tumors /Deaths	Tumor Incidence (%)	Tumors /Survivals	Tumor Incidence (%)	Tumors	Tumor Incidence (%)
CEF	HN302	37	13/24	54.17	13/13	100.00	26/37	70.27
	Md5	36	18/29	62.07	7/7	100.00	25/36	69.44
HVT	HN302	36	2/4	50.00	20/32	62.50 ^ab^	22/36	61.11 ^b^
	Md5	35	3/8	37.50	8/27	29.63 ^ab^	11/35	31.43 ^b^
CVI988	HN302	37	8/12	66.67 ^b^	15/25	60.00 ^ab^	23/37	62.16 ^b^
	Md5	38	0/11	0.00 ^ab^	5/28	17.86 ^ab^	5/38	13.16 ^ab^

* Equal volume of mock CEF suspension serves as vaccine negative controls. ^a^ Indicates significant difference between HVT- or CVI988-vaccinated groups and CEF-mock groups challenged by the same virus. ^b^ Indicates significant difference between HNLH302 and Md5 viruses in the same vaccination group.

## Data Availability

Not applicable.

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
