# Peer review of "Pathogenicity and Pathotype Analysis of Henan Isolates of Marek’s Disease Virus Reveal Long-Term Circulation of Highly Virulent MDV Variant in China"

_viruses, 2022, doi:10.3390/v14081651_

Round 1

Reviewer 1 Report

Dear authors,

This manuscript was substantially improved and much more clearly arranged.  As concern my comments, MDV strain information was presented well and more informative.

This revised manuscript is acceptable.

Reviewer 2 Report

N/A

This manuscript is a resubmission of an earlier submission. The following is a list of the peer review reports and author responses from that submission.

Round 1

Reviewer 1 Report

 General comments

This is an interesting paper that documents the relatively high virulence of strain HN302.  While I believe there is strong suggestion that this is a highly virulent MDV strain, I am not convinced the strain is vv+MDV.  The background and methodology of pathotyping are not described in the introduction, but there are specific protocols and comparisons required that have not been met according to validated methods. Pathotyping alternatives, such as ‘best fit’ pathotyping, have been described and validated, but this paper doesn’t mention such alternatives (1,2).  Whatever the alternative, it must be validated before being used to determine pathotype.  You must know that your method will correctly pathotype known strains. This paper roughly follows ‘best fit’ pathotyping methodology but fails to meet the vv+MDV criteria, which requires significantly higher %MD in HVT-vaccinated chickens compared to Md5.  This criteria is not met in Table 3. Using CVI988 to determine pathotype may be worthwhile, but previously has never been validated as an alternative method.  For ‘best fit’ pathotyping it is critical to compare unknown strain results to a prototype vv+MDV isolate, such as 648A. The authors should acquire this strain and compare against HN302. Table 4 appears to just be fishing for significant differences by separating out tumour incidence in deaths vs. survivals. Replicates are also critical in MDV pathotyping. So often there is variation from trial to trial, which might especially be the case in this trial without using defined inbred lines.  Including a second replicate would make the conclusions of this work much more convincing.

Specific comments

 Line 3: What is meant in the title by vv+MDV variant?  Does this mean it’s not a typical vv+MDV?

 Line 49: All viruses listed in this section should use alphaherpesvirus notation, for example Gallid alphaherpesvirus 2.

 Line 51: turkey should not be capitalized

 Line 71: Sentence is unclear. Emergence some of such viruses?

 Line 75: The introduction provides no guidance on established pathotyping methodology, and limitations associated with this study.

 Line 85: What is meant by variant strain?

 Line 98: Original reference for Md5 should be provided. Md5 was not isolated by Prof. Cui.

 Line 100: Merial doesn’t exist anymore.  Current company name should be listed. How many passages were the vaccines isolates (passages after isolation from the commercial vaccine)?

 Line 104: Has any testing been conducted to confirm purity of MDV isolates?  Co-infection with CIAV or other virus could cause false high virulence.

 Line 109: Organization of experiments is confusing.  It appears there were multiple animal experiments, as suggested by section 2.5, so this should all be outlined in section 2.3. 

 Line 137: Please justify challenge at 7 days of age.  This differs from established pathotyping methods.

 Line 148: Mortality without neurological lesions or tumors were considered MD positive?

 Line 180: Please include how many early non-specific deaths were removed from further analysis.

 Line 181: This is speculation.  What evidence do you have that early non-specific death was due to adverse effects from intra-abdominal infection?  Other laboratories use this same inoculation route without adverse effects.

 Line 190: What was typical tumor tropism of HN302?

 Line 212: How were 75-day and 91-day experiment durations determined?

 Line 316: This language is too strong.  This is not convincing evidence that HN302 is vv+MDV strain. It may be suggestive, but not convincing.

 Line 322: All Rispens vaccines are not equal.  This must be acknowledged when discussing lack of protection against your strains as you used a strain with lower protection compared to other commercial strains.

 Line 330: Please discuss specifics on Meq mutations of these isolates compared to previous papers that relate Meq mutations to virulence.

References cited above:

 1. Witter RL, Calnek BW, Buscaglia C, Gimeno IM, Schat KA. Classification of Marek's disease viruses according to pathotype: philosophy and methodology. Avian Pathol. 34:75-90; 2005 Apr.

2. Dunn JR, Reddy SM, Niikura M, Nair V, Fulton JE, Cheng HH. Evaluation and Identification of Marek's Disease Virus BAC Clones as Standardized Reagents for Research. Avian diseases. 61:107-114; 2017.

  1.  

Reviewer 2 Report

In the current manuscript titled “Pathogenicity and pathotype analysis of Henan isolates of Marek’s disease virus reveal long term circulation of vv+MDV variant in China”, Teng et al have isolated four MDV variants (HN502, HN302, HN304 and HN101) and present the pathogenicity and virulence of four Henan MDV isolates.

 The study is hypothesis driven with logically designed experiments. It is of general interest to the field of MDV vaccine developmental processes. It is scientifically sound and generally well presented to read, but had some concerns that the information presented did not present good enough for following/reading the author’s concept.

 Among four MDV isolates, HN302 could be a vv+MDV variant. MDV is an immunosuppressive and oncogenic virus. Meq is an oncogene of MDV and dispensable for cytolytic infection since MDV could be replicated well in the lymphoid organs and feather follicular epithelium. To provide the clear evidence, the authors should provide the phylogenetic analysis data of the sequence of the Meq gene. This data could tell us more convincing information whether four isolates (especially, HN302) are distinct from conventional MDV strains.

Reviewer 3 Report

This manuscript describes comparison of four Chinese MDV isolates for their virulence.  One isolate, HN302, demonstrated comparable MD incidence to a known very virulent MDV (vvMDV) strain, Md5.  This strain showed statistically significant reduction in the protection index (PI) in CVI988-vaccinated birds compared to Md5.  Also, the tumor occurances in the vaccinated birds were higher with HN302 compared to Md5 both in HVT-and CVI988-vaccinated birds.  Based on these findings, the authors concluded HN302 strain is a very virulent-plus MDV (vv+MDV) strain, which was not reported in China before.  

Major criticism

Although the experiments were planned and conducted properly, the conclusion that the isolate HN302 is a vv+MDV is misleading and inappropriate.  The MDV pathotypes such as vv- and vv+MDV were defined in very specific ways as proposed by Witter (reviewed in the reference 4 in this manuscript).  The vv+MDV was initially defined by the poor protection by bivalent vaccination with type 2 and type 3 virus vaccines.  In the reference 4, the authors stated “… it does not appear that the use of CVI988-vaccinated chickens will assist discrimination between the vv and vv+ pathotypes. However, CVI988-vaccinated chickens might be good models to detect new, but as yet unrecognized, pathotypes” since “the mean per cent MD lesions and the per cent protection between the two groups (vvMDV and vv+MDV) did not differ in CVI988-vaccinated chickens even though robust differences were noted in bivalent-vaccinated chickens.”  Though HN302 may represent a new pathotype, the authors can’t say whether HN302 may or may not fit to the published categorization of vv+MDV without using bivalently vaccinated birds according to this definition.  Please see Table 2 in the reference 4 carefully.  Some vvMDV were poorly protected by CVI988, while most of vv+MDV protected well by CVI988 vaccination.  Therefore, if the authors wish to argue HN302 is different and more difficult to protect birds by CVI988 vaccination than Md5 they should propose a separate term from vv+MDV to describe such MDV isolates to avoid confusion.

Minor points

Line 127: How did the authors select birds randomly for the weighing?  Weren’t MD affected-birds already apparently smaller at 21 dpi?

Lines 167-168: The authors should use more standard wording such as statistical difference instead of “an extremely remarkable difference”.

Table 1 (in the title line): Isn’t the host species for all Hyline brown, Huangshan yellow and Mayu chickens Gallus gallus?  A better tile is needed.

Lines 218-223: The authors stated “no significant difference of the cumulative morbidity, mortality or gross tumor incidences was observed…” in line 219 while “the HN302-challenged birds displayed a higher survival rate and …” in line 222.  Do the authors think the mortalities in unvaccinated birds different or similar?

Line 348, typo: widely should read wide.